# Modeling and Analysis of Foot Function in Human Gait Using a Two-Degrees-of-Freedom Inverted Pendulum Model with an Arced Foot

**DOI:** 10.3390/bioengineering10121344

**Published:** 2023-11-22

**Authors:** Qian Xiang, Shijie Guo, Jiaxin Wang, Kazunobu Hashimoto, Yong Liu, Lei Liu

**Affiliations:** 1Engineering Research Center of the Ministry of Education for Intelligent Rehabilitation Equipment and Detection Technologies, Hebei University of Technology, Tianjin 300401, China; 201811201005@stu.hebut.edu.cn (Q.X.); wangjx@hebut.edu.cn (J.W.); 202131205130@stu.hebut.edu.cn (Y.L.); 202131205073@stu.hebut.edu.cn (L.L.); 2The Hebei Key Laboratory of Robot Sensing and Human-Robot Interaction, Hebei University of Technology, Tianjin 300401, China; kazu_h@muf.biglobe.ne.jp; 3School of Mechanical Engineering, Hebei University of Technology, Tianjin 300401, China

**Keywords:** human gait, walking model, inverted pendulum model, roll factor

## Abstract

Gait models are important for the design and control of lower limb exoskeletons. The inverted pendulum model has advantages in simplicity and computational efficiency, but it also has the limitations of oversimplification and lack of realism. This paper proposes a two-degrees-of-freedom (DOF) inverted pendulum walking model by considering the knee joints for describing the characteristics of human gait. A new parameter, roll factor, is defined to express foot function in the model, and the relationships between the roll factor and gait parameters are investigated. Experiments were conducted to verify the model by testing seven healthy adults at different walking speeds. The results demonstrate that the roll factor has a strong relationship with other gait kinematics parameters, so it can be used as a simple parameter for expressing gait kinematics. In addition, the roll factor can be used to identify walking styles with high accuracy, including small broken step walking at 99.57%, inefficient walking at 98.14%, and normal walking at 99.43%.

## 1. Introduction

Walking is an important mode of transportation for people [1,2,3,4,5,6]. Thus, many kinds of lower limb exoskeletons that can assist in people’s walking have been developed [7,8,9,10,11,12,13,14]. However, the design of the exoskeletons mostly depends on the experience of engineers due to the lack of a reasonable gait model of a human body that can be used to analyze the gait characteristics of the coupled system composed of the exoskeleton and the wearer [15,16,17,18].

Modeling human walking is complex because it is a system with multiple joints, involving multiple muscles with different functions, as well as intermittent impulsive contact with the environment. The research on the kinematics and dynamics of human gait can be divided into two categories according to the different applications of the model: biomechanics gait analysis and robotics analysis [19]. The former typically uses a musculoskeletal model to investigate the details of human gait physiology, involving a large number of variables [20,21,22,23]. The latter analyzes human gait from a mechanical perspective, treating the human body as a rigid body, in order to establish a model that can be used for exoskeleton design [24,25].

Based on the energy conversion between dynamic potential energy during walking, human gait is usually approximated as a gait with an inverted pendulum model. Cavagna et al. [26] first proposed a one-degree-of-freedom (DOF) inverted pendulum model. The 1-DOF inverted pendulum is the simplest mechanical model and has been expanded in many versions (by adding springs, dampers, and telescopic actuators) [27,28,29,30,31,32,33,34,35,36] as the research foundation, making the model more accurate in expressing human walking. The most representative one is the 1-DOF inverted pendulum proposed by Gard et al. [37] based on a rocker foot, a human wheel-like gait can be characterized by the roll factor. This model is widely used in the study of leg mechanics [38,39,40,41,42]. However, the disadvantages of these models are the same as those of the 1-DOF inverted pendulum model, which is simple and difficult to generate a natural and realistic gait.

In order to better reflect human walking, more segments and joints have been added to the model. The model with multiple degrees of freedom has been extensively studied, which are multi-link models [43,44,45,46,47]. In the earliest of these, Hanavan et al. proposed a mathematical model with 15 links connected via spherical joints [44], but it is too complicated for practical use. Although both Hurmuzlu et al. [45], Ishigaki et al. [46], and Borisov et al. [47] have made models physiologically much closer to real human walking, they still cannot avoid complex calculations. High complexity and too many calculation parameters increase the calculation time and lead to poor comfort of the exoskeleton.

This paper establishes a kinematic model with simple parameters that can effectively express human walking characteristics for the design and motion planning of lower-limb-assisted exoskeletons. A kinematic analysis is conducted on the model, and expressions for the model parameters are derived. The relationship between model parameters and important gait parameters is studied, and the correctness of the model is verified. High-precision recognition of walking mode is performed using the model parameters. Finally, the application of the model parameters in the design of lower-limb-assisted exoskeletons is introduced.

## 2. Methods

### 2.1. Gait Model

Gard’s inverted pendulum model [37] is shown in Figure 1A, in which a virtual leg with length $L_{V}$ is introduced, and the foot is modeled as a rocker with radius r. The ratio of the length of the virtual leg to that of the real leg is called the roll factor and can be expressed as
(1)$$\rho_{G}=1/\left(1-r/L\right)$$
where $\rho_{G}$ represents the roll factor of the rocker-based inverted pendulum model proposed by Gard, and L represents the length of the leg. However, Gard’s model ignores the knee joints, making it unsuitable for the design of lower limb exoskeletons.

Considering the importance of the knee joint and the arced foot in human walking, we expand Gard’s model to a 2-DOF model as shown in Figure 1B. The human foot structure has the structure suitable for bipedal walking. As shown in Figure 1C, the ankle joint is located approximately a quarter of the foot length from the heel [48,49]. Humans can progress forward effectively by using the foot functions of heel-rocker, ankle-rocker, forefoot-rocker, and toe-rocker, which produce a wheel-like rolling motion under the foot [50]. Based on the foot’s structure and function, we model the foot as an unsymmetrical rocker to make the model more practical than Gard’s model (see Figure 1B). The center of mass (COM) is regarded as the intersection point of the two legs. From the kinematic analysis of the model, the roll factor can be expressed as
(2)$$\rho =1+f/\left(S_{l}-f\right)$$
where f represents the foot length, and $S_{l}$ represents the step length (see Figure 1D).

### 2.2. Gait Parameters

The parameters commonly used in gait analysis include step length, stride length, stride frequency, walking speed, gait cycle, gait phase, as well as the vertical excursion Δh of the COM [28,51]. From Equation (2), the step length can be expressed as
(3)$$S_{l}=\rho f/\left(\rho -1\right)$$

Considering a single support leg with an arced foot and a massless swing leg as shown in Figure 2, the moment balance centering at the ground contact point (GCP) (indicated by Q in Figure 2) is given with
(4)$$\frac{d\tau }{dt}=r_{com}\times F_{com}$$
where τ represents the rotational momentum around point Q, and $r_{com}$ and $F_{com}$ represent the vector from point Q to the COM and the inertial force acting on the COM, respectively. In a stable gait during the stance phase, since the angular momentum is conserved, Equation (4) equals 0, and the equation of motion can be expressed as
(5)$$m(g+{\ddot{y}}_{com})(x_{com}-x_{a})=m{\ddot{x}}_{com}(y_{com}-y_{a})$$
where $x_{com}$ and $y_{com}$ represent the position of COM in the x and y directions, g is the acceleration of gravity, and m represents the mass of the human body. Geometrically, $x_{com}-x_{a}$ and $y_{com}-y_{a}$ can be expressed as
(6)$$x_{com}-x_{a}≅-L_{iv}\theta_{r}{}^{\prime }-r(\theta_{r}-\beta_{r})$$
(7)$$y_{com}-y_{a}=L_{iv}\cos\theta_{r}{}^{\prime }≅L_{iv}$$
where $\theta_{r}{}^{\prime }$ represents the angle of the line of PQ from the vertical direction, $L_{iv}$ represents the distance between point P and point Q, and $\theta_{r}$ and $\beta_{r}$ represent the hip angle and the knee angle of the leading leg, respectively. The relationship between $\theta_{r}{}^{\prime }$ and $\theta_{r}$ is $\theta_{r}{}^{\prime }≅\frac{L}{\rho L_{iv}}\theta_{r}+\frac{l_{2}-r}{L_{iv}}\beta_{r}$. Ignoring that $\beta_{r}$ as the knee angle in the single leg support phase is small, Equation (5) can be simplified as
(8)$${\ddot{\theta }}_{r}≅\frac{g}{L}\left(2-\rho \right)\theta_{r}$$

The natural frequency of the 2-DOF inverted pendulum model is $\omega =\sqrt{\left(2-\rho \right)g/L}$, so the swing period can be expressed as $T=\sqrt{L/g}/\sqrt{2-\rho }$. The cadence (the number of steps per minute) $N_{C}$ is given with
(9)$$N_{C}=\left(1-2\xi \right)\sqrt{2-\rho }/T_{0}$$
where ξ represents the percentage of double support in a gait cycle, which is approximately 10% in normal stable walking, and $T_{0}$ is the inherent period of the inverted pendulum, where $T_{0}=\sqrt{L/g}$.

According to Equations (3) and (9), the gait speed can be expressed as
(10)$$V=\frac{\left(1-2\xi \right)f}{T_{0}}\frac{\rho \sqrt{2-\rho }}{\rho -1}$$

The vertical excursion of COM (Δh) is given with
(11)$$\Delta h=L-h=L-\left[l_{1}\cos\theta +(l_{2}-r)\cos(\theta -\beta )+r\right]$$
where h represents the vertical height of the COM. According to Equation (2), Equation (11) can be expressed as
(12)$$\Delta h=\rho f^{2}/\left[8L{\left(\rho -1\right)}^{2}\right]$$

As shown in Equations (3), (9), (10) and (12), the gait kinematics parameters are functions of the leg length L, the foot length f, and the roll factor ρ. Since the leg length and foot length are constant, the gait kinematics parameters are essentially functions of the roll factor ρ.

As shown in Figure 3, a gait cycle can be divided into seven phases: loading response (LR), mid-stance (MSt), terminal stance (TSt), pre-swing (PSw), initial swing (ISw), mid-swing (MSw), and terminal swing (TSw). Hip motion plays an important role when walking forward. In this figure, Hmax represents the highest point of forward flexion of the thigh, and Hmin represents the highest point of backward extension of the thigh. The two points give the thigh span during walking, which is strongly related to the step length. Kmax1 and Kmax2 represent the knee angle to achieve foot clearance at the initial swing phase and the maximum knee flexion angle, respectively. Amax1 represents the dorsiflexion angle at the heel strike, Amin1, the plantar flexion angle at the flat foot, Amax2, the dorsiflexion angle at the heel-off, and Amin2, the plantar flexion angle at the toe-off.

### 2.3. Gait Recognition

In normal walking, the step length is always larger than the foot length. Therefore, the minimum value of the roll factor is greater than 1. On the other hand, we understand from Equation (4) that the maximum value of the roll factor is less than 2 for an inverted pendulum motion. This is described in Figure 4. An abnormal gait with a roll factor of less than 1 usually occurs in the elderly with weakened lower limb muscle strength. When the roll factor is larger than 2, the gait is inefficient. Only when the value of the roll factor is between 1 and 2 does the human body walk efficiently and stably as an inverted pendulum. Therefore, we can use the roll factor to judge the walking state of a person.

## 3. Experiment

### 3.1. Subjects

Seven healthy adult subjects participated in this study (age = 27.8 ± 1.7 years, height = 169.8 ± 3.4 cm, and weight = 69.9 ± 5.3 kg). Subjects were healthy and regularly participated in moderate activity. They were free of any physical condition or limitation that prevented them from walking on a treadmill.

The test was approved by the ethics committee of Hebei University of Technology, and each subject read and provided written informed consent before the test. (Ethics committee name: the Biomedical Ethics Committee of Hebei University of Technology; approval code: HEBUThMEC2023017.)

### 3.2. Protocol

Each subject wore specially designed sportswear with reflective markers pasted on it. To perform the motion capture, reflective markers were attached to specific anatomical areas of the lower limbs according to the plug-in gait market set [52,53]. Figure 5A presents the locations of the markers that were pasted symmetrically from left to right, and 39 markers in total were pasted onto each subject. Each subject participated in two experiments, a treadmill test for model verification and a level walk test for gait recognition, under the VICON Motion System (Oxford Metrics Limited, Oxford, UK).

#### 3.2.1. Treadmill Test for Model Verification

Treadmill tests were conducted at 6 different speeds: 3.5, 4.0, 4.5, 5.0, 5.5, and 6.0 km/h. The test time at each speed was set to be 6 min, and each subject was asked to take a rest of 10 min to ensure physical recovery after the 6 min test at a certain speed (see Figure 5C). The first 3 min of the 6 min test was for warming up to make the subject adapt to the treadmill walk. As shown in Figure 5C, the walking speed in the test was changed from slow to fast and then from fast to slow.

#### 3.2.2. Level Walk Test for Gait Recognition with the 2-DOF Model

Level walk tests were conducted to investigate the usefulness of the proposed 2-DOF model in identifying different gaits. Each subject was asked to walk on the ground in three patterns for 100 steps, without any restriction on walking speed. The three patterns were abnormal small broken steps, inefficient walking, as well as normal walking. Herein inefficient walking means very slow walking with little ankle joint motion.

### 3.3. Data Collection and Kinematic Parameter Calculation

Data were collected at 100 Hz using the VICON Motion system (Oxford Metrics, Oxford, UK) with 10 cameras (model: VANTAGE-V5-VS-5299). The real marker trajectory data were filtered with a quintic spline filter based on code written by Herman Woltring before the modeling stage [54].

The gait cycle was calculated by taking the time difference between two consecutive heel landings of the left foot. Regarding how to recognize landing, we judged the heel landing of a foot when the marker at the heel of that foot reached its lowest position during walking. The stride was defined as the horizontal distance in the sagittal between the two heel landings, while the step length was half of the stride. The cadence was obtained by counting the number of steps in the test time (three minutes) and dividing the number by three. The vertical position of the COM and the angles of the hip, knee, and ankle joints were obtained by using the plug-in gait model in Vicon Nexus software v1.8.5. The plug-in gait model is a commonly used version of the conventional gait analysis models [55,56,57]. The output angles for all joints were calculated from the YXZ Cardan angles derived by comparing the relative orientations of the two segments [52].

The roll factor in the proposed 2-DOF inverted pendulum model was calculated using Equation (2), using the information of step length and foot length. The foot length of each subject was measured directly.

Using Vicon Polygon (Oxford Metrics Group, Oxford, UK) [52], kinematic data were extracted to a C3D file or ASCII file, which was then placed in MATLAB software (MATLAB 23.2.0) for post-processing.

The recognition accuracy is the walking state that correctly determines this walking state. And the error rate is other types of walking states mistaken for this walking state. The recognition accuracy and the error rate were calculated using MATLAB software to calculate the roll factor values for each step of each subject in different walking states.

### 3.4. Statistical Analysis

Statistical analysis was performed using the SPSS statistical software system (SPSS Inc., Chicago, IL, USA; version 22.0). Means and standard deviations for each test condition were calculated. One-way repeated measures analyses of variance with six conditions (six walking speeds) were used to verify the effect of the roll factor on step length, cadence, gait speed, and the vertical excursion of the COM, as well as the angles of the hip, knee, and ankle joints. Pairwise comparisons with Bonferroni post hoc tests were conducted to identify differences between conditions when a statistically significant main effect was identified with the one-way repeated measures analyses of variance. A paired *t*-test was performed to assess the difference between the roll factor and the kinematic parameters. *p* < 0.05 represented a significant difference. Linear regression and curve-fitting methods were used to fit the measurement curves of the kinematic parameters. Linear regression analyses were performed to calculate the slope of the relationship between the important characteristic points of the hip, knee, and ankle joint angles and the roll factor. The formula for calculating goodness of fit was $R^{2}=\mathrm{ESS}/\mathrm{TSS}=1-\mathrm{RSS}/\mathrm{TSS}$. To determine the correlation strength between the calculation using the proposed 2-DOF model and the measurement, the 2-DOF model calculation and the measurement were compared using Pearson’s product-moment correlation coefficients.

## 4. Results

### 4.1. Gait Analysis

The kinematic parameters of the seven subjects in the treadmill test are plotted against roll factors in Figure 6 for both measurements and calculations using the proposed 2-DOF model. The black fitting curves of the gait parameter were obtained using the measured values (black points) that were acquired from able-bodied subjects (the goodness-of-fit values were $R^{2}=0.997$, $R^{2}=0.972$, $R^{2}=0.998$, and $R^{2}=0.980$). The red fitting curves of the gait parameter were obtained using the 2-DOF inverted pendulum walk model (the goodness-of-fit values were $R^{2}=0.997$, $R^{2}=0.998$, $R^{2}=0.995$, and $R^{2}=0.998$). The model well agrees with the measurements, demonstrating the validity of the proposed model. The Pearson correlation between the model and the measurements were 0.996, 0.995, 0.998, and 0.994 for step length, cadence, gait speed, and vertical excursion of the COM, respectively (*p* < 0.01).

Figure 6 also demonstrates that the roll factor is a comprehensive parameter expressing gait kinematic parameters, including step length, cadence, gait speed, vertical excursion of the COM, etc. During normal walking, the roll factor ranged from 1.4 to 1.8, the vertical excursion of the COM ranged from 3 to 8 cm, and the value was 5 cm at the preferred gait speed.

The hip joint angles at different speeds are shown in Figure 7A. It shows that the Hmax increases with the increase in speed, although there are individual differences between different subjects. From Figure 7B,C, we know that both Hmin and Hmax have a linear relationship with the roll factor (the Hmax goodness-of-fit values are $R^{2}=0.940$, $R^{2}=0.948$, $R^{2}=0.984$, and $R^{2}=0.920$; the Hmin goodness-of-fit values are $R^{2}=0.959$, $R^{2}=0.986$, $R^{2}=0.971$, and $R^{2}=0.975$). As shown in Figure 7D, the difference between Hmax and Hmin also decreases linearly with the roll factor (the goodness-of-fit values are $R^{2}=0.943$, $R^{2}=0.942$, $R^{2}=0.991$, and $R^{2}=0.956$).

As shown in Figure 8, the knee joint angles increase with the gait speed, but both Kmax1 and Kmax2 decrease linearly with the roll factor (the Kmax1 goodness-of-fit values are $R^{2}=0.973$, $R^{2}=0.952$, $R^{2}=0.939$, and $R^{2}=0.984$; the Kmax2 goodness-of-fit values are $R^{2}=0.944$, $R^{2}=0.987$, $R^{2}=0.982$, and $R^{2}=0.992$). This figure also shows that the roll factor can be used as a factor to express knee motions during walking.

Figure 9 shows that with the decrease in the roll factor, the dorsiflexion angle at heel strike (Amax1) and the plantar flexion angle at flat foot (Amin1) decrease, while the plantar flexion angle at the toe-off (Amin2) increases in the negative direction in a linear way (Figure 9B,C,E), and the dorsiflexion angle at the heel-off (Amax2) increases linearly with the roll factor (Figure 9D). The dorsiflexion angle at the heel strike (the Amax1 goodness-of-fit values are $R^{2}=0.970$, $R^{2}=0.938$, $R^{2}=0.951$, and $R^{2}=0.990$) and the plantar flexion angle at the toe-off (the Amin2 goodness-of-fit values are $R^{2}=0.942$, $R^{2}=0.953$, and $R^{2}=0.967$) have high correlation with the roll factor (Figure 8B,E). In contrast, the Amin1 (Figure 9C) and Amax2 (Figure 9D) show low goodness-of-fit values (the Amin1 goodness-of-fit values are $R^{2}=0.411$, $R^{2}=0.425$, $R^{2}=0.733$, and $R^{2}=0.527$; the Amax2 goodness-of-fit values are $R^{2}=0.006$, $R^{2}=0.112$, $R^{2}=0.668$, and $R^{2}=0.613$). This means the two values have no relationship with the roll factor.

In summary, the roll factor of the 2-DOF inverted pendulum model has a similar inverse proportion function to the gait parameters. The 2-DOF model can express the foot rocker function from heel-rocker to toe-rocker via the roll factor.

### 4.2. Gait Recognition

Since the roll factor has a strong relationship with the kinematics parameters, as shown in the above figures, and that its value is related to the walking style, as shown in Figure 4, we have an idea that uses the roll factor to identify walking styles. The accuracy of recognizing gait according to the value of the roll factor is given in Table 1. The average recognition accuracy for the three typical walking styles of seven subjects is 99.57%, 98.14%, and 99.43%. The results demonstrate that that the roll factor can be used as a parameter to identify walking styles.

## 5. Discussion

The important characteristic points of hip joint angles are Hmin and Hmax. When the Hmin feature points appear, the extension movement of the thigh reaches its maximum, and the hip flexor muscles are mainly activated, with the peak flexion torque reaching its maximum. When the hip joint rapidly flexes, there is an energy burst in the sagittal plane. At the Hmax feature point, the thigh flexes to its maximum motion, providing assurance for a sufficient step. The flexion and extension of the hip joint help advance the legs forward while maintaining balance in the body. Hmin and Hmax have a linear relationship with the roll factor (Figure 7). The roll factor can be used as an evaluation of thigh extension and flexion. This is consistent with the fact that gait may be improved through the modification of the foot rocker shape proposed by Gard et al. [37].

Knee flexion (Kmax1) provides shock absorption when a human foot follows the ground. The shock absorption is of great significance to the stability of human walking. The larger the Kmax2 flexion angle, the easier to complete the clearance of the foot contour. The effective completion of the flexion angle of the knee joint at the initial swing phase can avoid tripping over obstacles slightly higher than the ground during walking. The roll factor has a linear relationship with the flexion of the knee (Figure 8). The roll factor can be used to judge stable walking.

The four rolling functions of the foot on the ground correspond to the important characteristic points of the ankle joint angles (Amax1: heel-rocker, Amin1: ankle-rocker, Amin2: forefoot-rocker, and Amax2: toe-rocker). The push-off of the trailing leg and the heel collision of the leading leg during the double support phase are important. It can be seen in Figure 9B that the smaller the roll factor, the more obvious the heel-rocker, and the more advantageous the human gait. Human plantigrade gait combined with heel strike appears to be an adaptation for aerobics, long-distance travel, and the effective energetic costs of locomotion. We suggest the continuous use of heel-rocker walking and evaluate it with the roll factor. Figure 9E shows that the toe-rocker becomes more obvious as the roll factor decreases. The ankle joint produces the highest mechanical power with the toe-rocker, with the peak being more than three times the maximum power produced by the other joints [58]. In Figure 8, the 2-DOF model reflects the foot rocker function from heel-rocker to toe-rocker through the roll factor.

The internal factor that causes changes in the kinematic characteristic parameters of the lower limbs is the change in muscle strength. The weakening of lower limb muscle strength greatly reduces the movement of the hip, knee, and ankle joints [59,60,61]. The roll factor has a high linear correlation with the characteristic points of these angles, indicating that it can reflect the intensity of lower limb muscle strength.

Our study found that the roll factor can not only serve as a criterion for evaluating walking ability, but also as a parameter for adjusting the assist curve of the lower limb assist exoskeleton.

In Equation (2), the step length divided by foot length is a dimensionless ratio. This dimensionless ratio is a directly proportional function of $\rho /\left(\rho -1\right)$ (the goodness-of-fit value is $R^{2}=0.998$) (Figure 10A). Therefore, the step length is an amplification of the foot function via the roll factor. In the range of 3.5 to 6.0 km/h of walking speed, the dimensionless ratio is 2.33 to 3.48, and the longer the step length, the better the foot rolling function. Human walking involves an energy exchange between gravitational potential energy (position energy) and forward kinetic energy (motion energy). During stable walking, when the human body is in the highest vertical position, the forward speed is the lowest, and when the human body is in the lowest vertical position, the forward speed is the highest. The total mechanical energy of the human body, i.e., the sum of gravitational potential and kinetic energy, is almost constant. The vertical excursion of the COM reflects the energy conversion in the process of human walking. As shown in Figure 6D, in the range of 1.4–1.7, the smaller the roll factor, the greater the vertical excursion of the COM, and the more gravitational potential energy is converted into kinetic energy. Therefore, the energy conversion during walking can be analyzed with the roll factor, i.e., a longer effective leg length is energetically advantageous [62]. The step length has a trade-off relation with the effective leg length, and the roll factor has an optimal value for low energy consumption during walking (Figure 10B).

The optimal value of the roll factor for an efficient walk depends on the walking style. Human walking tends to choose a step length or step frequency that minimizes metabolic energy consumption at a given walking speed [63]. It can be seen in Figure 7, Figure 8 and Figure 9 that at the same speed, the roll factor value of subject 4 is generally higher than that of the other three subjects. A small step length is a combination of a small hip rotation (Figure 7D), large knee flexion (Figure 8B), small ankle flexion (Figure 9B), and small ankle extension (Figure 9E). This means that as the walking speed increases, subject 4 mainly reduces the energy consumption by increasing the stride frequency.

The roll factor can be used to identify walking styles with high accuracy. Therefore, when wearing the lower-limb-assisted exoskeleton for walking, different assistance plans are carried out in different walking states. When the wearer walks with a small broken step, the exoskeleton does not provide assistance. When the wearer is in an inefficient walking state, the exoskeleton continues to increase assistance until the wearer enters an efficient walking state. After the wearer reaches an efficient walking state, the exoskeleton adjusts the assist function based on the optimal value of the roll factor to provide power that maintains high efficiency and energy conservation. In addition, with the development of rocker sole shoes [64,65], the roll factor can also be used as a method to select rocker shoes that are suitable for efficient and energy-saving walking.

## 6. Conclusions

This paper proposed a 2-DOF inverted pendulum walk model and defined a new parameter, roll factor, for expressing gait styles. The kinematics of human gait were investigated using this parameter. It was demonstrated that the roll factor has a strong relationship with other gait kinematics parameters, so it can be used as a simple parameter for expressing gait kinematics. In addition, the roll factor can be used to identify walking styles with high accuracy, at 99.57% for small broken step walking, at 98.14% for inefficient walking, and at 99.43% for normal walking. The roll factor can be a criterion for evaluating walking ability. In addition, it can also be a parameter for adjusting the assist function of the lower-limb-assisted exoskeleton. We will try to introduce the proposed 2-DOF inverted pendulum walk model and the defined roll factor into the design of lower limb exoskeletons in future work.

## Figures and Tables

**Figure 1 bioengineering-10-01344-f001:**
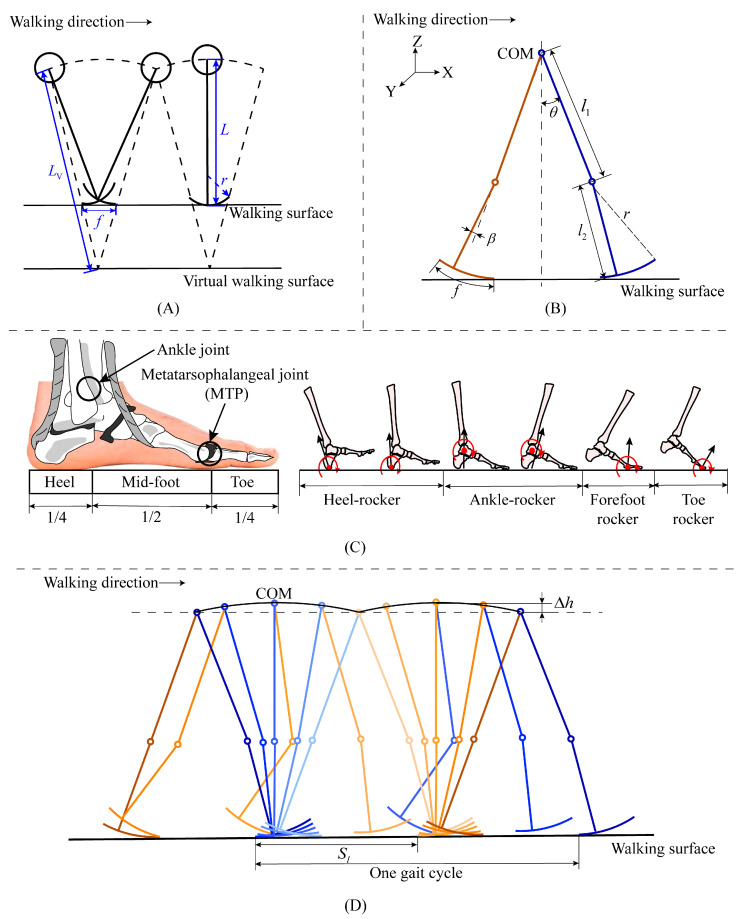
Rocker-based inverted pendulum model of human walk. (**A**): Gard’s 1-DOF model, (**B**): the proposed 2-DOF model, (**C**): human foot structure and function, and (**D**): description of step length.

**Figure 2 bioengineering-10-01344-f002:**
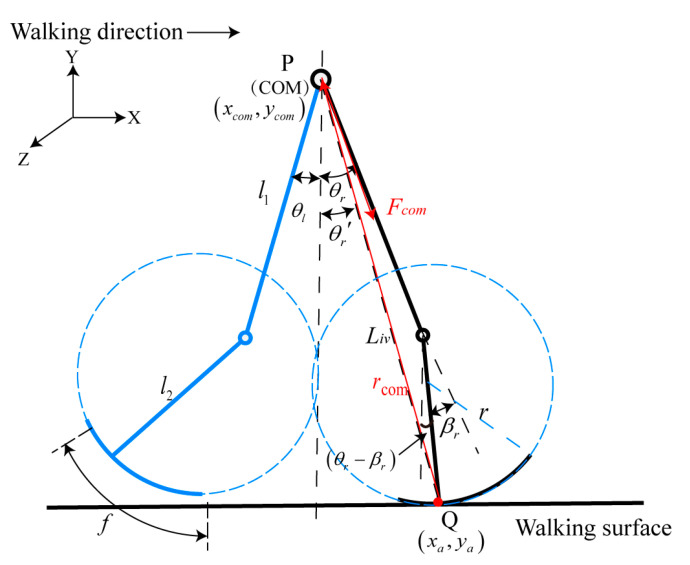
The moment balance centering on the point Q for the 2-DOF model.

**Figure 3 bioengineering-10-01344-f003:**
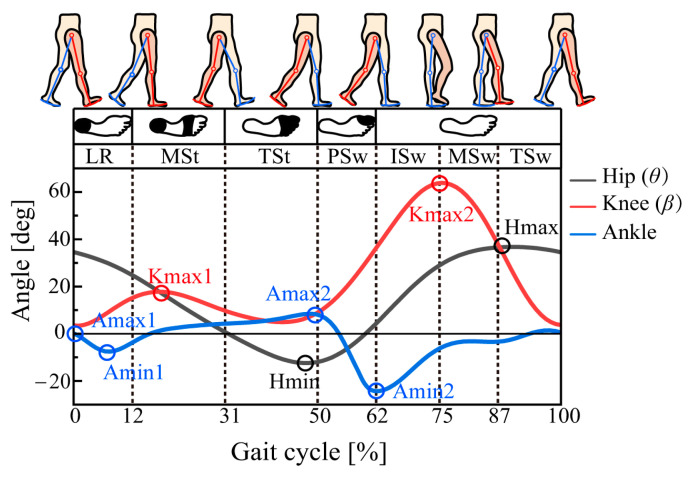
Gait phase division and the important characteristic points of hip, knee, and ankle joint angles.

**Figure 4 bioengineering-10-01344-f004:**
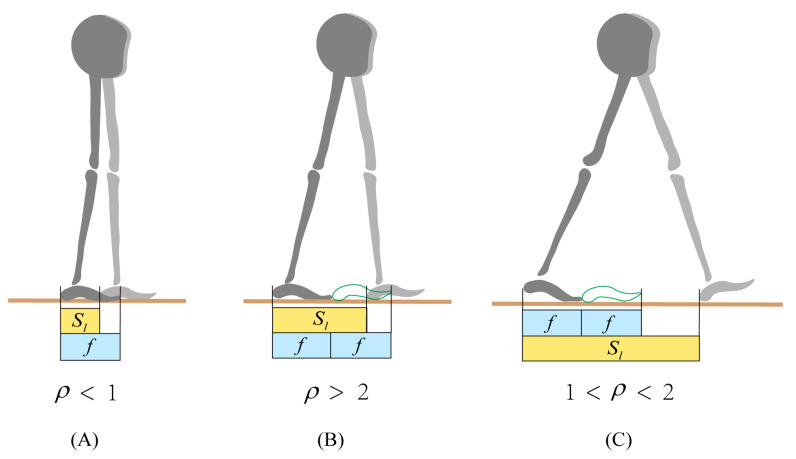
Walking state and roll factor. (**A**): Small broken step walking, (**B**): inefficient walking, and (**C**): inverted pendulum walking.

**Figure 5 bioengineering-10-01344-f005:**
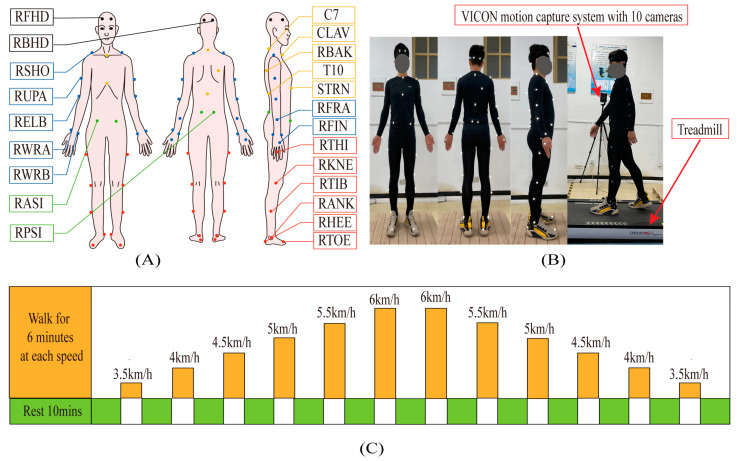
(**A**,**B**): Locations of reflective markers for motion capture, including 4 markers on the head (the black points); 5 markers on the trunk (the yellow points) at the 7th cervical vertebra, the 10th lumbar vertebra, the upper end of the sternum stem, the lower end of the sternum stem, and the middle of the right scapula; 14 markers on the upper limbs (the blue points, 7 on each side); 4 markers on the pelvis (the green points) at the left and right anterior superior iliac spine and left and right posterior superior iliac spine; and 12 markers on the lower limbs (the red points, 6 on each lower limb). (**C**): Test process.

**Figure 6 bioengineering-10-01344-f006:**
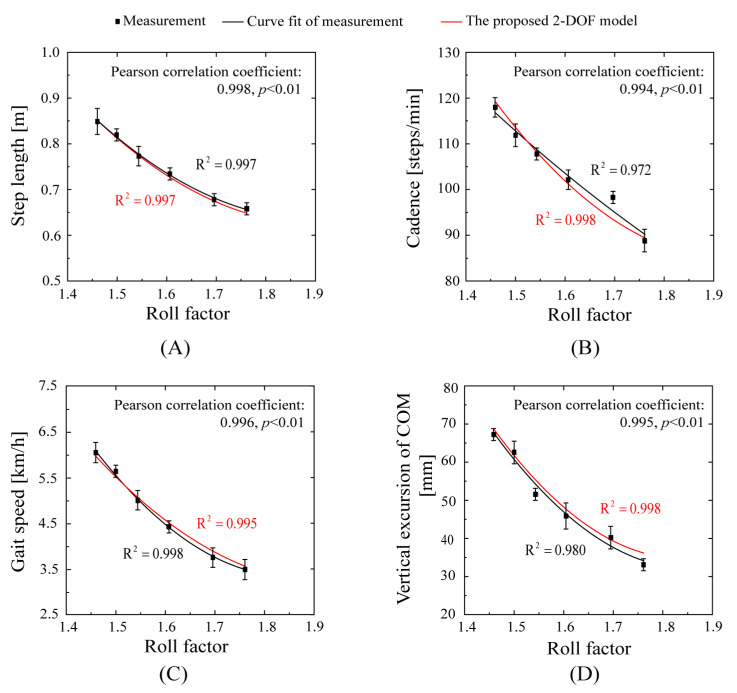
Gait kinematic parameters versus roll factor. (**A**): step length; (**B**): cadence; (**C**): gait speed; (**D**): vertical excursion of COM.

**Figure 7 bioengineering-10-01344-f007:**
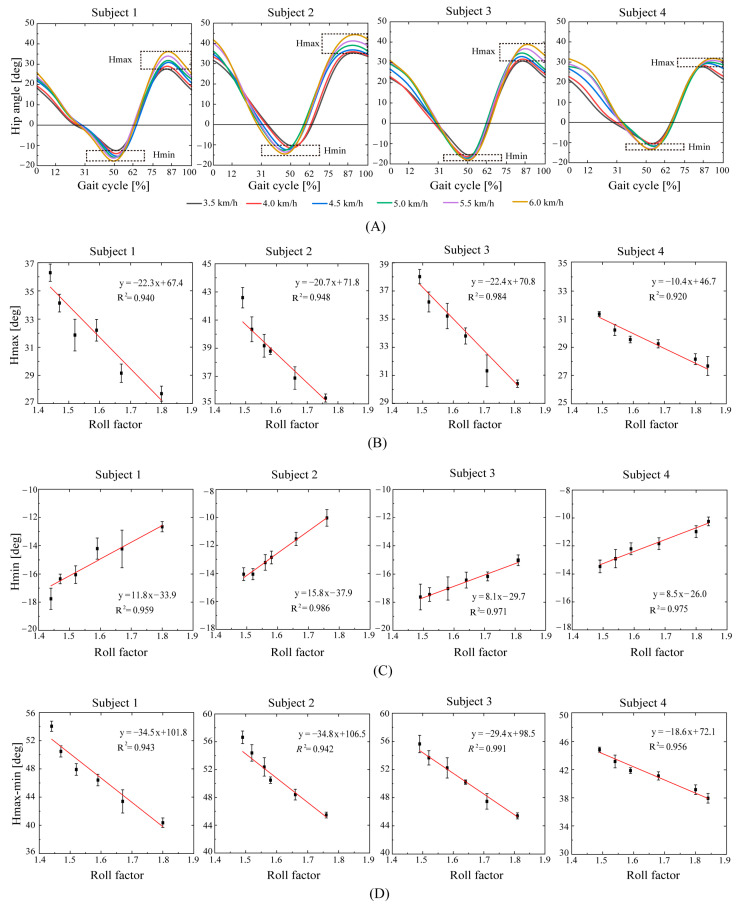
(**A**): The relationship between hip joint angles and roll factor. (**B**): Hmax: the highest point of forward flexion of the thigh; (**C**): Hmin: the highest point of backward extension of the thigh gait; (**D**): the thigh span during walking.

**Figure 8 bioengineering-10-01344-f008:**
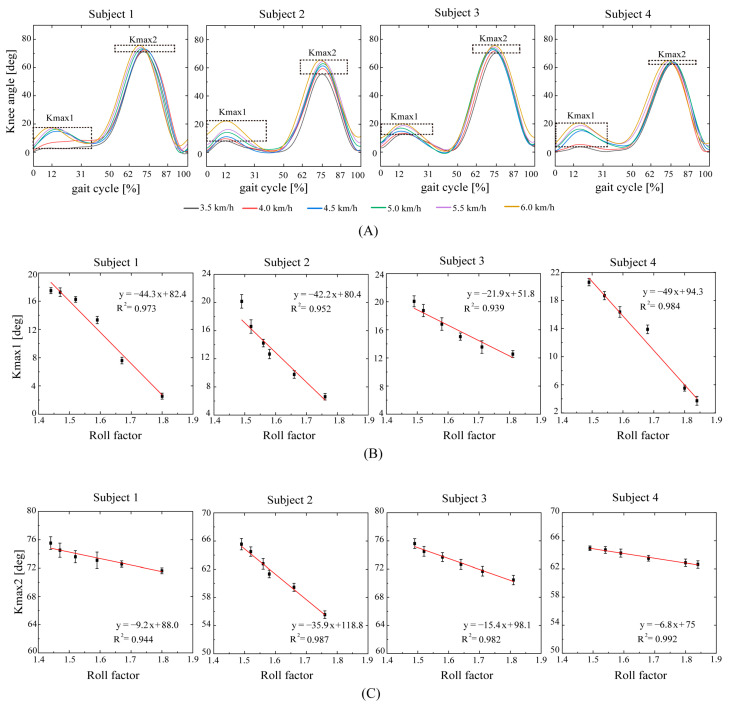
(**A**): The relationship between knee joint angles and the roll factor. (**B**): Kmax1: the maximum knee flexion angle; (**C**): Kmax2: the knee angle to achieve foot clearance at the initial swing phase.

**Figure 9 bioengineering-10-01344-f009:**
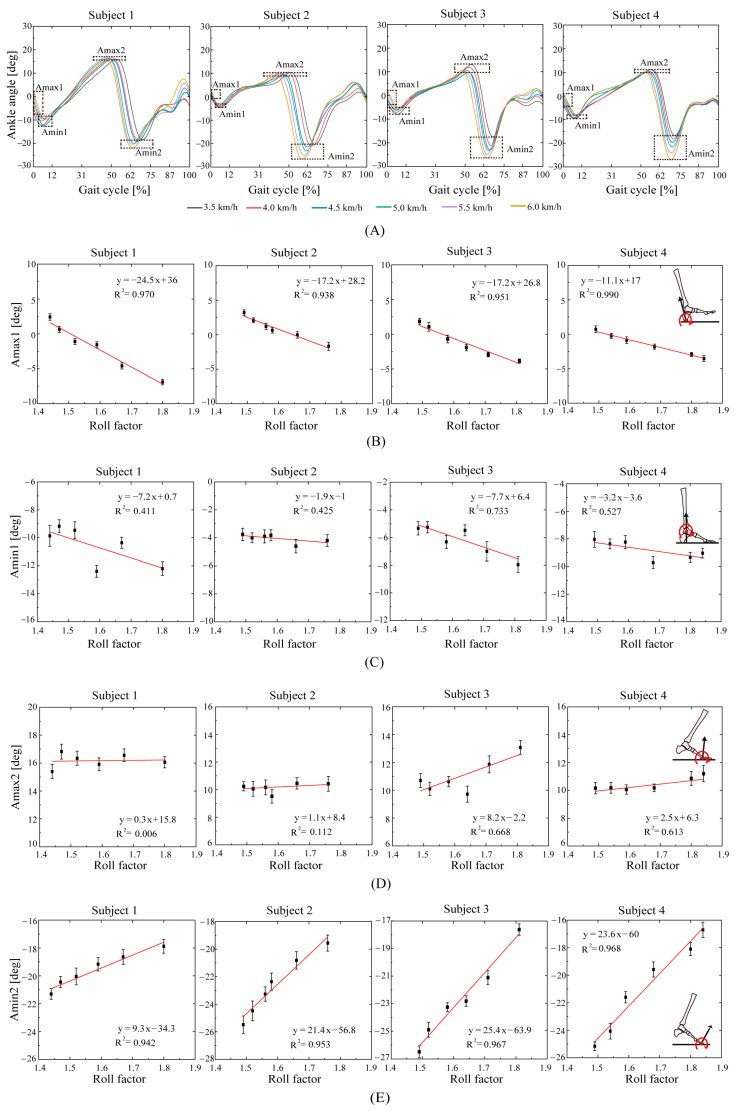
(**A**): The relationship between ankle joint angles and the roll factor. (**B**): Amax1: the dorsiflexion angle at the heel strike; (**C**): Amin1: the plantar flexion angle at the flat foot; (**D**): Amax2: the dorsiflexion angle at the heel-off; (**E**): Amin2: the plantar flexion angle at the toe-off.

**Figure 10 bioengineering-10-01344-f010:**
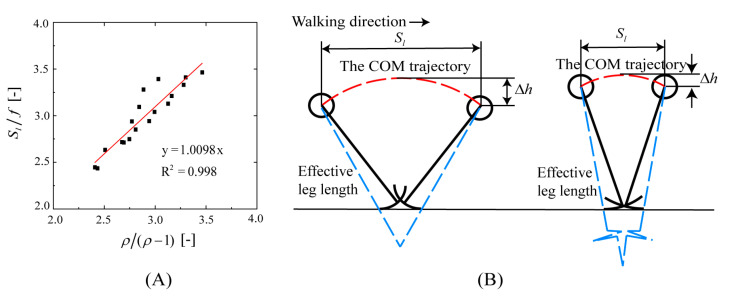
(**A**): The relationship between the dimensionless ratio and $\rho /\left(\rho -1\right)$; (**B**): the trade-off relationship between step length and effective leg length.

**Table 1 bioengineering-10-01344-t001:** Recognition accuracy of the three typical walking styles.

Types	Small Broken Step Walking		Inefficient Walking		Inverted Pendulum Walking	
Subject Number	Accuracy ^1^ (%)	Error Rate ^2^ (%)	Accuracy (%)	Error Rate (%)	Accuracy (%)	Error Rate (%)
No. 1	100	0	98	0	100	1
No. 2	100	0	100	0	100	0
No. 3	99	0	98	0.5	100	1
No. 4	99	0	100	2.5	96	0
No. 5	100	0	97	0	100	1.5
No. 6	99	0	98	0.5	100	1
No. 7	100	0	96	0	100	2
Average	99.57	-	98.14	-	99.43	-

^1^ The accuracy is the walking state correctly determined as this walking state. ^2^ The error rate is other types of walking states mistaken for this walking state.

## Data Availability

The original data are available following reasonable request.

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
