# Peer review of "Modeling and Analysis of Foot Function in Human Gait Using a Two-Degrees-of-Freedom Inverted Pendulum Model with an Arced Foot"

_bioengineering, 2023, doi:10.3390/bioengineering10121344_

Round 1

Reviewer 1 Report

Comments and Suggestions for Authors

Manuscript title: Modeling and Analysis of Foot Function in Human Gait Using a 2-DOF Inverted Pendulum Model with an Arc Foot

Journal: Bioengineering

This paper proposes a 2-DOF inverted pendulum walk model and defines a new parameter, roll factor, for expressing gait styles. The kinematics of human gait are investigated by the parameter. It is demonstrated that the roll factor has a strong relationship with other gait kinematics parameters, so it can be used as a simple parameter for expressing gait kinematics. I support this work to be published in the journal Bioengineering. However, a revision may be needed to further improve the scientific findings.

1.    The authors modified Eq. 1 to fit for an unsymmetrical rocker because human foot is not symmetric around the ankle joint. However, the detailed basis of the modification are not clear.

2.    In Table 1, the authors should clarify how they calculated the error rate.

Author Response

Dear Reviewer:

Thank you for your review of our paper.

Due to your suggestions, the revised manuscript has become better and readers can obtain more valuable information.

We have answered each of your points below.

Kind regards,

Shijie Guo, Dr.

Reviewer 2 Report

Comments and Suggestions for Authors

This study examined modeling and analysis of foot function in human gait using a 2-dof inverted pendulum model with an arc foot. This study is intriguing and decently executed; however, some important improvements are needed regarding the writing and explaining methods. In particular, statistical analyses and data analysis, as well as the participants' number. More specific comments are the following:

Title: Please avoid using abbreviations in the title of the study.  

Introduction:

1.     The first paragraph of the Introduction is only focusing on the elderly. I believe others can also benefit from exoskeletons, like people with disabilities or with MS or Cerebral palsy. Therefore, don`t limit this only to the elderly and add additional references regarding those above.

2.     Lines 38-39: Avoid mentioning methods in the introduction.

3.     Paragraphs 3 and 4 are simply poorly written. Authors should tell us a story instead of merely tossing one research after another. Please rewrite these paragraphs to improve readability.

4.     Please clearly write the aim of the study and the proposed hypotheses.

Methods:

1.     Line 164: Please add the number of the Ethical approval.

2.     What was the criterion for selecting just 7 participants? Is it enough to reach the conclusions? Please elaborate.

3.     Please cite the relevant literature regarding the positioning of the reflective markers.

4.     Same with the treadmill protocol. Please cite the relevant literature utilizing this protocol.

5.     For data collection, please indicate how you analyze the data. Software, filters used…

6.     Please include the statistical analysis paragraph explaining all the statistical procedures used.

Discussion

1.     The discussion is very brief and does not elaborate on all of the most important results. Please improve your discussion to be less repeating the results and more explaining them, elaborating…

Conclusions:

1. It would be beneficial for this paper to add some practical implications of these results.

Author Response

Dear Reviewer:

Thank you very much for taking the time to read and revise our manuscript.

Thank you for your valuable suggestions. You have made comprehensive corrections to the structure, content, research methods, results, etc. of our manuscript, which will play a very important role in improving the quality of the manuscript.

We have carefully answered your comments one by one and made careful modifications. Please see the attachment.

Round 2

Reviewer 2 Report

Comments and Suggestions for Authors

Dear authors,

Thank you for the effort to improve the manuscript. I`m please with the answers and resolved issues.

Best regards